

# Adaptive sentiment analysis using multioutput classification: a performance comparison

Taqwa Hariguna[1] and  Athapol Ruangkanjanases[2]

[1] Information Systems, Universitas Amikom Purwokerto, Purwokerto, Jawa Tengah, Indonesia
[2] Department of Commerce Chulalongkorn Business School, Chulalongkorn University, Bangkok, Thailand

## ABSTRACT

The primary objective of this research is to create a multi-output classification model for sentiment analysis through the combination of 10 algorithms: BernoulliNB, Decision Tree, K-nearest neighbor, Logistic Regression, LinearSVC, Bagging, Stacking, Random Forest, AdaBoost, and ExtraTrees. In doing so, we aim to identify the optimal algorithm performance and role within the model. The data utilized in this study is derived from customer reviews of cryptocurrencies in Indonesia. Our results indicate that LinearSVC and Stacking exhibit a high accuracy (90%) compared to the other eight algorithms. The resulting multi-output model demonstrates an average accuracy of 88%, which can be considered satisfactory. This research endeavors to innovate in adaptive sentiment analysis classification by developing a multi-output model that utilizes a combination of 10 classification algorithms.

## INTRODUCTION

One method for increasing the accuracy of sentiment analysis is the development of multi-output models (*Wu et al., 2019*; *Alfazzi, 2022*). Sentiment analysis, a subfield of natural language processing (NLP), seeks to extract and interpret the emotions or attitudes conveyed in a text. It is commonly utilized in marketing, public opinion, and social media (*Kabir et al., 2021*).

There are various methods for determining the sentiment of a text, including the use of multi-output models. A multi-output model is capable of predicting multiple output classes simultaneously. In the context of sentiment analysis, such models can be utilized to predict the overall sentiment of a text and identify the specific type of sentiment present (*e.g.*, positive, negative, or neutral) (*Kaur et al., 2021*; *Hayadi et al., 2023*).

The significance of developing multi-output models for sentiment analysis lies in their ability to enhance the accuracy of sentiment analysis. Single-output models, which can only predict a single output class (*e.g.*, only positive or negative), may struggle to detect more subtle forms of sentiment, such as irony or sarcasm (*Han, Liu & Jin, 2020*). By contrast, multi-output models can more accurately identify the specific type of sentiment in a text,

Corresponding authors
Taqwa Hariguna,
taqwa@amikompurwokerto.ac.id
Athapol Ruangkanjanases,
athapol@cbs.chula.ac.th

enabling more informed decision-making based on the sentiment conveyed (*Sun et al., 2019*; *Saputra, Rahayu & Hariguna, 2023*; *Cao, Chen & Liu, 2023*). In sentiment analysis, multiple outputs refer to a situation where a model must output multiple decisions or labels for each input. A combination of algorithms is often employed in various studies to improve the accuracy of multi-output models in sentiment analysis.

The ensambling method is a commonly utilized approach in multiple-outcome sentiment analysis research. This method involves the creation of multiple models using different algorithms, with the decisions of each model serving as new features for a final model (model c) of the same type. The final model can then learn from the decisions of the previous models and adapt accordingly, leading to improved accuracy.

*Han, Liu & Jin (2020)* utilized sentiment analysis and machine learning techniques in conjunction with one another to predict stock market movements. The algorithms employed included support vector machines (SVM) and Naive Bayes. Previous research has demonstrated that this combination of algorithms exhibits higher accuracy than using a single algorithm. In a study conducted by *Sun et al. (2019)*, an ensemble algorithm consisting of multiple algorithms was utilized to perform sentiment analysis on social media data. The algorithms employed included Random Forest, Decision Tree, and Naive Bayes, and the results indicated that combining these algorithms was more effective than using a single algorithm.

*Mukhtar, Khan & Chiragh (2018)* combined deep learning and feature selection techniques to perform sentiment analysis, employing algorithms such as convolutional neural networks (CNN) and logistic regression. The results indicated that the combination of these algorithms exhibited higher accuracy than using a single algorithm. Similarly, in a study by *Miao et al. (2021)*, feature selection techniques and classification algorithms were combined to perform sentiment analysis, including recursive feature elimination (RFE) and SVM. The results demonstrated that this combination of algorithms had higher accuracy than using a single algorithm.

*Salazar, Montoya-Múnera & Aguilar (2022)* combined deep learning techniques with rule-based approaches to perform sentiment analysis on Twitter data, utilizing algorithms such as long short-term memory (LSTM) and decision tree. The results indicated that the combination of these algorithms exhibited higher accuracy than using a single algorithm.

The innovative aspect of this research lies in the combination of 10 algorithms in a multi-output classification model, a practice that is uncommon in various studies, particularly in the field of sentiment analysis. The primary objectives of this study are to demonstrate the effectiveness of the combination of 10 algorithms and to compare the performance of the individual algorithms to identify the optimal algorithm.

## MATERIALS & METHODS

Figure 1 illustrates the process of developing the sentiment data classification model used in this study:

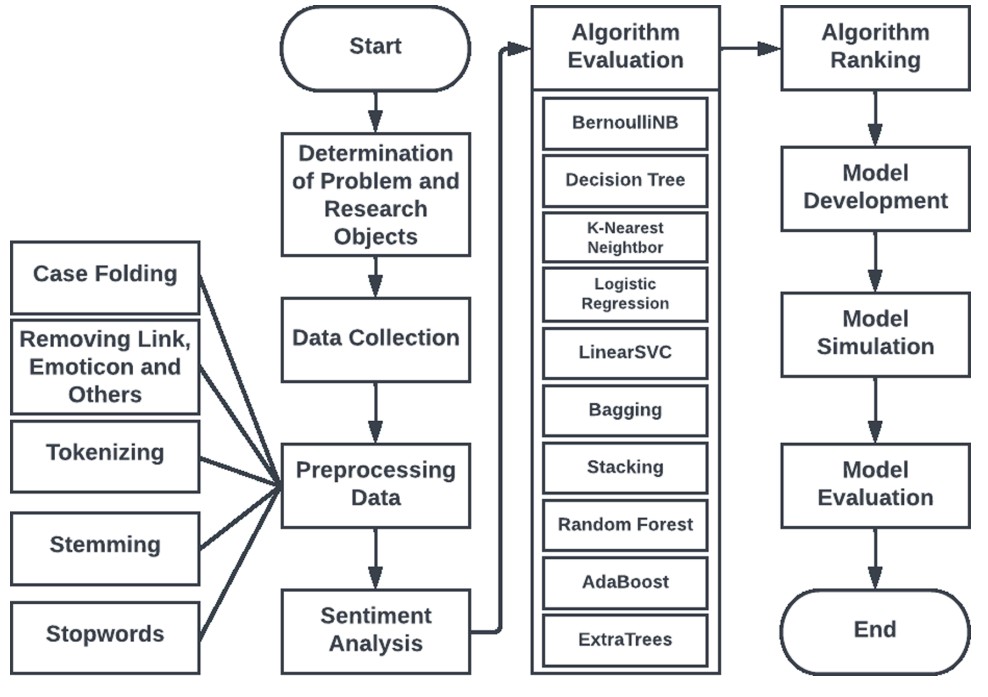

**Figure 1** Research steps.

## Identification of the problem and research objectives

The aim of this research is to develop a model capable of performing sentiment analysis through the combination of 10 algorithms: BernoulliNB, Decision Tree, K-Nearest Neighbor, Logistic Regression, LinearSVC (Support Vector Classification), Bagging, Stacking, Random Forest, AdaBoost, and ExtraTrees. Once the model has been developed, a comparison will be conducted to identify the algorithm with the highest level of performance in sentiment analysis. The findings of this study will aid in the improvement of the model's ability to perform sentiment analysis and provide insight into the strengths and limitations of each algorithm utilized.

## Data collection

The data collection method used in this study is data collection from platforms that provide customer reviews on two cryptocurrency exchanges in Indonesia: Indodax and Binance. Data collection was performed by accessing the customer reviews available on the platform. The amount of data collected is 1,599 data points that have been prepared in terms of characteristics and normalization. The data collected is considered valid and relevant for this research. In addition, the data also undergoes a normalization process to facilitate analysis and interpretation. Data normalization is carried out with the aim of eliminating scale differences between data and equalizing the magnitude of data values in one particular attribute. This is important because data that has differences in scale can interfere with the accuracy of the analysis and interpretation results. By using data that has been prepared in terms of characteristics and normalization, this research can facilitate the

analysis and interpretation process and produce more accurate results (*Abo et al., 2018*; *Amalia, Gunawan & Nasution, 2021*; *Yadav & Vishwakarma, 2020*; *Tseng, Liang & Tsai, 2022*; *Ajiono & Hariguna, 2023*; *Zeng & Nantida, 2023*).

## Preprocessing

The data in this study underwent several preprocessing steps to prepare them for further analysis. The preprocessing phase included several procedures, including capitalization, punctuation and conjunction removal, tokenization, stemming, and stop word elimination. Capitalization was used to remove all distinctions between uppercase and lowercase letters irrelevant to the meaning of the word. The next step was to remove punctuation and hyphens as they did not provide useful information for text analysis. Tokenization was used to divide the text into tokens, which could include words, phrases, or sentences with different meanings.

Stemming is a natural language processing technique used to extract the root or base form of a word by removing any prefixes or suffixes attached to the word. The goal of stemming is to reduce each word to a common base form, even if it is not a valid English word, to group words that have the same meaning. By doing so, stemming can help to reduce the dimensionality of text data and simplify analysis. Rather, converting a word to its base form is known as lemmatization (*Del Arco et al., 2022*; *Glimsdal & Granmo, 2021*; *Greco & Polli, 2020*; *Suhane & Kowshik, 2021*; *Jen & Lin, 2021*; *Prayitno, Saputra & Waluyo, 2021*). Stemming, however, involves reducing the inflected form of a word to its root. In addition, stop words, which are common English words such as "the," "a," "an," and "is," were removed during the preprocessing phase because they do not provide significant meaning in text analysis. The purpose of these preprocessing steps was to prepare the data for further analysis and facilitate data analysis.

## Model development

After the data is available, the next step is to select the algorithm to be incorporated into the model (*Budi & Yaniasih, 2022*; *Jafarian et al., 2021*; *Kaur et al., 2021*). In general, machine learning algorithms such as Naive Bayes, SVM, and neural networks are utilized in sentiment analysis. However, depending on the purpose and characteristics of the data used, other algorithms such as rule-based algorithms or hybrid algorithms may be employed.

The next step after choosing an algorithm is to implement it in the model being developed. This typically involves converting the data into a format that is suitable for the selected algorithm and adjusting any necessary parameters (*Xu et al., 2019*; *Liu, 2023*; *Wang, Sungho & Cattareeya, 2023*). After implementing the algorithm, the model must be validated by comparing the results it produces with the actual results (ground truth). This allows for the accuracy of the model to be determined. Upon validating the model, it is necessary to optimize it in order to obtain improved results. The optimization can be achieved through a variety of methods, such as altering the model's hyperparameters, modifying the number of layers or neurons, or selecting more effective features.

# RESULTS

In this results section, the research results are presented in accordance with the research objectives, specifically the comparison of the performance of the algorithms used and the results of the developed model.

## Algorithm comparation

The accuracy of an algorithm refers to the degree of accuracy it achieves in classifying data. A higher accuracy indicates a better performance of the algorithm in classifying data.

The BernoulliNB algorithm, one of the naive Bayes algorithms used for binary classification, achieved an accuracy of 0.87, indicating that it is able to classify data with 87% accuracy. The DecisionTreeClassifier algorithm, a decision tree algorithm used to classify data based on its attributes, achieved an accuracy of 0.86, indicating that it is able to classify data with 86% accuracy.

The KNeighborsClassifier algorithm, a k-nearest neighbor algorithm used to classify data based on the nearest data in the neighborhood, achieved an accuracy of 0.87, indicating that it is able to classify data with 87% accuracy. The LogisticRegression algorithm, a logistic regression algorithm used to classify binary data, achieved an accuracy of 0.89, indicating that it is able to classify data with 89% accuracy. The LinearSVC algorithm, a support vector machine algorithm that utilizes a linear kernel to classify data, achieved an accuracy of 0.90, indicating that it is able to classify data with 90% accuracy.

The BaggingClassifier algorithm, a bagging algorithm that improves the accuracy of the algorithm by using multiple models simultaneously, achieved an accuracy of 0.89, indicating that it is able to classify data with an accuracy of 89%. The StackingClassifier model achieved an accuracy of 0.90, indicating that it is able to classify data into the appropriate class with an accuracy of 90%.

The RandomForestClassifier model achieved an accuracy of 0.88, indicating that it is able to classify data into the correct class with an accuracy of 88%. The AdaBoostClassifier model also achieved an accuracy of 0.89, indicating that it is able to classify data into the correct class with an accuracy of 89%. Finally, the ExtraTreesClassifier model achieved an accuracy of 0.88, indicating that it is able to classify data into the correct class with an accuracy of 88%. The performance results of each algorithm can be seen in Fig. 2.

Linear Support Vector Classification (LinearSVC) and Stacking Classifier are two algorithms that have demonstrated high accuracy in the classification of sentiment in text data. In particular, both algorithms have shown to be effective in the detection and analysis of the sentiment (positive or negative) present in a given text. In comparison to other algorithms, LinearSVC and Stacking Classifier have demonstrated superior performance in the task of sentiment analysis.

Linear Support Vector Classification (LinearSVC) utilizes the Support Vector Machine method to predict the class of an input data point through the identification of the optimal hyperplane that separates two or more classes. This algorithm is highly accurate and has been shown to be effective for both binary (two classes) and multi-class (more than two classes) classification tasks. Its performance has been extensively demonstrated in various

| Model | Accuracy | Precision | Recall | ROC Score |
|---|---|---|---|---|
| BernoulliNB | 0.87 | 0.72 | 0.68 | 0.53 |
| DecisionTreeClassifier | 0.86 | 0.88 | 0.70 | 0.60 |
| KNeighborsClassifier | 0.87 | 0.64 | 0.71 | 0.61 |
| LogisticRegression | 0.89 | 0.70 | 0.61 | 0.60 |
| LinearSVC | 0.90 | 0.72 | 0.88 | 0.63 |
| BaggingClassifier | 0.89 | 0.65 | 0.76 | 0.62 |
| StackingClassifier | 0.90 | 0.68 | 0.77 | 0.69 |
| RandomForestClassifier | 0.88 | 1.00 | 0.83 | 0.52 |
| AdaBoostClassifier | 0.89 | 0.57 | 0.45 | 0.70 |
| ExtraTreesClassifier | 0.88 | 0.66 | 0.76 | 0.50 |

**Figure 2  Algorithm rankings.**

studies and has established itself as a reliable and effective method for sentiment analysis and other classification tasks.

StackingClassifier is an ensambling method that combines multiple machine learning models into a more powerful model. StackingClassifier predicts the class of an instance using multiple machine learning models as base models, and then combines the prediction results of the base models with other machine learning models as metamodels. In this way, StackingClassifier can improve classification accuracy by using multiple machine learning models, each of which has different advantages and disadvantages.

A multiple-output classification accuracy of over 80% can be deemed satisfactory, especially when multiple algorithms are utilized. However, it is important to note that this benchmark may vary depending on the specific context being analyzed and should not be treated as a definitive metric of an algorithm's superiority. Additionally, it is crucial to consider the benefits and drawbacks of each algorithm in relation to the problem at hand, as no algorithm is superior in all scenarios.

## Model simulation and evaluation

In order to utilize the model, the input data must be entered in the form of sentiments or ratings for analysis. The data is then processed through the PreProcessed Review process, which displays the analyzed data. Subsequently, the Predictions By Classifiers component utilizes 10 algorithms from the multi-output model to analyze the data. The results of this analysis are presented on a scale from 0 to 10, with 0 representing negative sentiment and 10 representing positive sentiment. Finally, the TextBlob Sentiment Polarity (UnPreprocessed)

```
In [933]: #Simulasi
          custom_review = "lumayan sih transaksi di triv tapi kyk ga ada yang pake ini website yah"
          custom_review

Out[933]: 'lumayan sih transaksi di triv tapi kyk ga ada yang pake ini website yah'

In [934]: generate_rating(ml_models_list,custom_review)

          PreProcessed Review:
          ['lumayan sih transaksi di triv tapi kyk ga ada yang pake ini website yah']

          Predictions By Classifiers:

          BernoulliNB : [0]
          DecisionTreeClassifier : [1]
          KNeighborsClassifier : [1]
          LogisticRegression : [1]
          LinearSVC : [0]
          BaggingClassifier : [0]
          StackingClassifier : [1]
          RandomForestClassifier : [1]
          AdaBoostClassifier : [0]
          ExtraTreesClassifier : [1]

          Dampak Komentar (0: Negative | 10: Positive): 5

          TextBlob Sentiment Polarity(UnPreprocessed): 0.5
```

**Figure 3** Model result towards neutral data.

display shows the number of words that are not recognized by the model. The visualization of this model usage process is illustrated in Fig. 3.

The average accuracy of the developed model is 88%, which is considered to be quite accurate as long as the input rating or sentiment data can be recognized by the model. However, if the rating or sentiment data is dominated by words that are unfamiliar to the model or have not been captured in its training data, most of the model's algorithms may classify the data as negative.

The results of the model are not simply binary (positive or negative) but rather are presented as a score from 1 to 10 in order to more accurately reflect the sentiment of the input data. This is because a given sentence may not contain exclusively positive or negative words, and presenting the results on a more nuanced scale allows for greater flexibility and validity in different analysis contexts. Furthermore, using a scale rather than a binary positive/negative categorization allows for a more granular and potentially more actionable measurement. Additional simulation results of the model can be found in Figs. 4 and 5 below.

## DISCUSSION

In recent years, sentiment analysis has become a widely researched topic owing to its potential application in various fields. In the current work, the effectiveness of LinearSVC and StackingClassifier algorithms in sentiment analysis was investigated, and their superior performance compared to other algorithms was demonstrated. This result is consistent with previous studies, such as the work of *Aini et al. (2021)*, which showed that SVM-based algorithms, such as LinearSVC, are effective in sentiment classification. Furthermore, the study highlights the importance of considering the specific context being analyzed when evaluating the performance of sentiment analysis algorithms. This finding is also supported by *Hananto, Rahayu & Hariguna (2022)*, who noted that the accuracy of sentiment analysis algorithms can vary depending on the type of data being analyzed and the specific domain

```
In [922]: custom_review = "Keren nih transaksi di indodax, bagus banget pelayanannya apalagi kalo ada member banyak deh promonya"
          custom_review

Out[922]: 'Keren nih transaksi di indodax, bagus banget pelayanannya apalagi kalo ada member banyak deh promonya'

In [923]: generate_rating(ml_models_list,custom_review)

          PreProcessed Review:
          ['Keren nih transaksi di indodax, bagus banget pelayanannya apalagi kalo ada member banyak deh promonya']

          Predictions By Classifiers:

          BernoulliNB : [1]
          DecisionTreeClassifier : [1]
          KNeighborsClassifier : [0]
          LogisticRegression : [1]
          LinearSVC : [1]
          BaggingClassifier : [1]
          StackingClassifier : [1]
          RandomForestClassifier : [1]
          AdaBoostClassifier : [1]
          ExtraTreesClassifier : [1]

          Dampak Komentar (0: Negative | 10: Positive): 9

          TextBlob Sentiment Polarity(UnPreprocessed): 0.0
```

**Figure 4** **Model result towards positive data.**

or application. Therefore, it is important to evaluate the performance of sentiment analysis algorithms in relevant contexts to ensure their effectiveness.

The use of ensemble methods, such as the StackingClassifier, has also been explored in sentiment analysis. In a study by *Hariguna & Rachmawati (2019)*, it was demonstrated that ensemble methods can improve sentiment classification accuracy by combining predictions from multiple base models. This finding is consistent with the results of the current study, which showed that the StackingClassifier can improve classification accuracy by leveraging the strengths and mitigating the weaknesses of multiple machine learning models. The current study also used a multi-output classification model to classify sentiments in tweets. This approach is similar to the work of *Hariguna & Rachmawati (2019)*, who used a multilabel classification method to classify sentiments in movie reviews. The use of multiple algorithms in the multi-output classification model used in the current study is innovative and represents a potential improvement over previous methods that used only a single algorithm.

Finally, the high accuracy of the multi-output classification model used in this study suggests its potential for real-world applications. This finding is consistent with the work of *Hung (2021)*, who demonstrated the usefulness of sentiment analysis in predicting stock market trends. The high accuracy of sentiment analysis algorithms, such as the multi-output classification model used in this study, can enable more accurate predictions and decision-making in various fields. In conclusion, the current study provides valuable insights into the effectiveness of sentiment analysis algorithms, particularly LinearSVC and StackingClassifier, in relevant contexts. The use of multioutput classification models and ensemble methods is also innovative and represents potential improvements over previous methods. The results of this study are consistent with previous research in this area and suggest the potential for real-world applications of sentiment analysis. Based on the obtained research results, the followings can be pointed out:

First, the results of this study demonstrated that LinearSVC and StackingClassifier are two algorithms that can be effectively utilized in sentiment analysis, or the process

```
In [931]:  custom_review = "Pengalaman terburuk yang saya dapat adalah ketika proses pengiriman dengan binance sangat mahal dan lama"
           custom_review

Out[931]:  'Pengalaman terburuk yang saya dapat adalah ketika proses pengiriman dengan binance sangat mahal dan lama'

In [932]:  generate_rating(ml_models_list,custom_review)

           PreProcessed Review:
           ['Pengalaman terburuk yang saya dapat adalah ketika proses pengiriman dengan binance sangat mahal dan lama']

           Predictions By Classifiers:

           BernoulliNB : [0]
           DecisionTreeClassifier : [0]
           KNeighborsClassifier : [0]
           LogisticRegression : [0]
           LinearSVC : [0]
           BaggingClassifier : [0]
           StackingClassifier : [0]
           RandomForestClassifier : [0]
           AdaBoostClassifier : [0]
           ExtraTreesClassifier : [0]

           Dampak Komentar (0: Negative | 10: Positive): 0

           TextBlob Sentiment Polarity(UnPreprocessed): 0.0
```

**Figure 5    Model result towards negative data.**

of identifying and analyzing the sentiments (positive or negative) expressed in a text. Both algorithms exhibited higher accuracy in classifying sentiments compared to other algorithms. LinearSVC utilizes the support vector machine (SVM) method to predict the class of an instance by finding the optimal hyperplane to differentiate between two classes. This algorithm is recognized for its high accuracy and strong performance in both binary and multi-class classification tasks.

In contrast, StackingClassifier is an ensemble method that combines multiple machine learning models to form a more robust model. It predicts the class of an instance by using multiple machine learning models as base models and then aggregating the prediction results of these base models with the use of other machine learning models as meta-models. As a result, StackingClassifier can improve classification accuracy by leveraging the strengths and mitigating the weaknesses of multiple machine learning models. A classification accuracy of over 80% for tasks with multiple output classes can be deemed satisfactory, especially when utilizing multiple algorithms. However, it is important to note that this benchmark may vary depending on the specific context being analyzed and should not be treated as a definitive measure of an algorithm's superiority. Additionally, it should be remembered that no algorithm is superior in all scenarios, so it is crucial to consider the benefits and drawbacks of each algorithm in relation to the problem at hand.

Secondly, the multi-output classification model used in this study demonstrated superiority due to its ability to classify data into multiple output classes. This is particularly useful for tasks that involve classifying data with multiple output variables, as was the case in this study, which utilized 10 algorithms to classify sentiment in tweets. Additionally, the multi-output classification model employed in this study represents a novel approach as it is the first multi-output sentiment classification model to utilize 10 algorithms. This is likely to be highly beneficial in improving the accuracy of the model, as the use of multiple algorithms provides a greater amount of information that can be utilized in the classification process.

Furthermore, the accuracy of the multiple algorithm classification model used in this study was found to be quite high at 88%. This indicates that the model is capable of accurately classifying data, making it reliable for identifying the sentiment of tweets. It can therefore be concluded that the multi-output classification model utilized in this study is a superior and innovative approach for classifying data, particularly in the context of sentiment research.

## CONCLUSIONS

This study successfully developed a sentiment classification model that combines 10 algorithms, resulting in an average accuracy of 88%. This approach demonstrates the potential of combining multiple classification algorithms to improve sentiment analysis accuracy. However, while the accuracy achieved in this study represents a significant increase compared with previous studies (*Han, Liu & Jin, 2020*; *Miao et al., 2021*; *Salazar, Montoya-Múnera & Aguilar, 2022*; *Sun et al., 2019*), there are still limitations to consider. One limitation is the potential bias in the data used to train the model because sentiment analysis is highly dependent on the context in which the text is produced. Therefore, future studies should focus on creating more diverse and representative training datasets to improve the accuracy of the model. Additionally, while LinearSVC and StackingClassifier showed high accuracy in sentiment classification, it is important to note that no algorithm was superior in all scenarios. Therefore, future research should focus on evaluating the advantages and disadvantages of other algorithms and how they can be combined to improve the accuracy.

Furthermore, the use of multiple algorithms for sentiment classification requires significant computational resources and time. Therefore, future research should focus on optimizing the model to reduce computational cost while maintaining accuracy. Overall, although the sentiment classification model developed in this study demonstrated significant improvements in accuracy compared to previous studies, there are still limitations and opportunities for future work. By addressing these limitations and exploring alternative approaches, the accuracy of sentiment classification models can be further improved, ultimately leading to more accurate and reliable sentiment analysis in various contexts.

Based on the conclusions of this study, there are several future perspectives and suggestions. First, although the developed sentiment classification model has been shown to be accurate, further studies should be conducted to improve its performance. This can be achieved by exploring the potential of combining additional classification algorithms to improve the accuracy of the model. Additionally, the model can be tested on a larger dataset to evaluate its scalability and robustness. Second, future studies can explore the use of deep learning techniques such as convolutional neural networks (CNNs) and recurrent neural networks (RNNs) for sentiment analysis. These techniques have shown promising results in natural language processing tasks and may provide a more accurate and efficient approach to sentiment analysis.

Third, future studies could investigate the applicability of the developed sentiment classification model to other domains, such as product reviews, movie reviews, and news

articles. This can provide insight into the generalizability of the model and its potential use in other contexts. Finally, future studies can explore the ethical implications of sentiment analysis and the use of machine learning algorithms for decision-making. This can include examining the issues of bias, fairness, and accountability in the development and use of sentiment analysis models. Finally, several avenues for future research and suggestions can be made based on the findings of this study. These include improving the accuracy of the developed sentiment classification model, exploring the use of deep-learning techniques, evaluating the generalizability of the model, and examining the ethical implications of sentiment analysis.

### Funding
The authors received no funding for this work.

### Competing Interests
he authors declare that they have no competing interests.

### Author Contributions
- Taqwa Hariguna conceived and designed the experiments, performed the experiments, analyzed the data, performed the computation work, prepared figures and/or tables, authored or reviewed drafts of the article, and approved the final draft.
- Athapol Ruangkanjanases conceived and designed the experiments, performed the experiments, analyzed the data, performed the computation work, prepared figures and/or tables, authored or reviewed drafts of the article, and approved the final draft.

### Data Availability
The raw measurements and code are available in the Supplementary File.

### Supplemental Information
Supplemental information for this article can be found online at http://dx.doi.org/10.7717/peerj-cs.1378#supplemental-information.

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
