# Peer review of "Adaptive sentiment analysis using multioutput classification: a performance comparison"

_PeerJ Computer Science, doi:10.7717/peerj-cs.1378_

## Round 0.1 · original submission · Major Revisions

Your manuscript “Adaptive sentiment analysis using multioutput classification: a performance comparison” has been assessed by our reviewers. They have raised a number of points which we believe would improve the manuscript and may allow a revised version to be published in PeerJ Computer Science. Please carefully consider the reviewers' concerns, as a few critical points were raised that need to be satisfyingly addressed to make the paper acceptable for publication.

·

Basic reporting

The paper is discretly written and clear in form even for those who are not experts in sentiment analysis.
It almost conforms to PeerJ standards, but I suggest the authors improve the structure by adding figures and tables before references.

Order and center the figures within the text.

Supplement figures with captions.

Revise the grammar better.

Experimental design

This study aims to innovate the adaptive classification of sentiment analysis by developing a multi-output model using a combination of 10 classification algorithms.
The proposed methodologies are rigorously defined, but I suggest the authors better discuss the various steps and add more information about the collected data.

In the part of the preprocessing there is a wrong definition.
Each word is not converted into its basic form (base word) in the stemming process.
The conversion of a word into its base form (known as "lemma") is the Lemmatization process, while Stemming is the process of reducing the inflected form of a word into its root.
Review this important step in your experimental analysis.

Validity of the findings

The results of this study demonstrated that LinearSVC and StackingClassifier are two algorithms that can be effectively utilized in sentiment analysis.

In the "Discussion" section, better describe what is innovative about your analysis compared to other similar studies conducted on the dataset of interest.

Conclusions should be better formulated, outlining the advantages and limitations of the approaches used and defining how the accuracy of other, less accurate classifiers can be improved in future work, while the accuracy of better classifiers can be increase by more.

Additional comments

The paper is accepted if revisions are made.

·

Basic reporting

The paper is written clearly and coherently, which makes a significant contribution to scientific information and research. Good use of the English language is highlighted.
In this paper, the authors present a a multi-output classification model for sentiment analysis through the combination of 10 algorithms: BernoulliNB, Decision Tree, K-nearest neighbor, Logistic Regression, LinearSVC, Bagging, Stacking, Random Forest, AdaBoost, and ExtraTrees.
The data utilized in this study is derived from customer reviews of cryptocurrencies in Indonesia. The results indicate that LinearSVC and Stacking exhibit a high accuracy (90%) compared to the other eight algorithms.
Authors are advised to insert the figures after the description.

Experimental design

Excellent title and abstracts that concretely define the general work and outline it in detail. The methodology used reflects clarity and good readability. The results are very interesting as regards the metrics related to the different classifiers.

Validity of the findings

The conclusions of the paper are well done, they contain all the work done, it is recommended only to the authors to extend 'future works'.

---

## Round 0.2 · Major Revisions

Your manuscript has been assessed by a reviewer. They have raised a number of points which we believe would improve the manuscript and may allow a revised version to be published in PeerJ. Please carefully consider the reviewers' concerns, as a few critical points were raised that need to be satisfyingly addressed to make the paper acceptable for publication.

·

Basic reporting

The paper is well written and clear in form even for those who are not experts in sentiment analysis.

The new version is conform to the PeerJ standard.

Figures with captions have been added, helping to improve the previous version of the document.

The idea of reporting parts of the code implementation makes your work more authentic.

Experimental design

This study aims to innovate the adaptive classification of sentiment analysis by developing a multi-output model using a combination of 10 classification algorithms.

The proposed methodologies are rigorously defined, but I suggest the authors to correct the following definition in the line 101:
"The stemming process aims to convert each word into its base form, which is the root word. This helps eliminate any variation in the word form that does not change the meaning of the word.", but pay attention!
The root is that element to which we usually add endings, and possibly prefixes or suffixes.
The basic form is the incoming word, our starting point, in processes such as derivation.

Therefore, if you transformer the word in the basic form the process isa called Lemmatization; if you transformer the word in its root, the process is called Stemming.

Review this important step in your experimental analysis.

Validity of the findings

The results of this study demonstrated that LinearSVC and StackingClassifier are two algorithms that can be effectively utilized in sentiment analysis.

The "Discussion" section has been expanded and considers what is innovative about the analysis compared to other similar studies conducted on the dataset of interest.

The Conclusions have been rephrased better, outlining the advantages and limitations of the approaches used and defining how the accuracy of the classifiers is better than other classification algorithms.

Additional comments

The paper is accepted if revisions are made, in particular those concerning the stemming/lemmatization phase in the preprocessing of the experimental phase.

---

## Round 0.3 · accepted · Accept

Based on my own assessment as Editor, I am pleased to inform you that the manuscript is acceptable for publication in PeerJ Computer Science.